# Total Saponins Isolated from Corni Fructus via Ultrasonic Microwave-Assisted Extraction Attenuate Diabetes in Mice

**DOI:** 10.3390/foods10030670

**Published:** 2021-03-22

**Authors:** Shujing An, Dou Niu, Ting Wang, Binkai Han, Changfen He, Xiaolin Yang, Haoqiang Sun, Ke Zhao, Jiefang Kang, Xiaochang Xue

**Affiliations:** 1National Engineering Laboratory for Resource Development of Endangered Crude Drugs in Northwest China, College of Life Sciences, Shaanxi Normal University, Xi’an 710119, China; Asjcyp@163.com (S.A.); kristin@snnu.edu.cn (D.N.); wangting86@snnu.edu.cn (T.W.); hanbinkaisnnu@163.com (B.H.); hcf14416106@163.com (C.H.); yangxiaolin163@126.com (X.Y.); sunhaoqiang32@163.com (H.S.); 2The Key Laboratory of Medicinal Resources and Natural Pharmaceutical Chemistry, The Ministry of Education, College of Life Sciences, Shaanxi Normal University, Xi’an 710119, China; zhaoke_1999@163.com

**Keywords:** *Cornus officinalis*, saponins, ultrasonic extraction, microwave extraction, response surface methodology, type 2 diabetes mellitus

## Abstract

Saponins have been extensively used in the food and pharmaceutical industries because of their potent bioactive and pharmacological functions including hypolipidemic, anti-inflammatory, expectorant, antiulcer and androgenic properties. A lot of saponins-containing foods are recommended as nutritional supplements for diabetic patients. As a medicine and food homologous material, Corni Fructus (CF) contains various active ingredients and has the effect of treating diabetes. However, whether and how CF saponins attenuate diabetes is still largely unknown. Here, we isolated total saponins from CF (TSCF) using ultrasonic microwave-assisted extraction combined with response surface methodology. The extract was further purified by a nonpolar copolymer styrene type macroporous resin (HPD-300), with the yield of TSCF elevated to 13.96 mg/g compared to 10.87 mg/g obtained via unassisted extraction. When used to treat high-fat diet and streptozotocin-induced diabetic mice, TSCF significantly improved the glucose and lipid metabolisms of T2DM mice. Additionally, TSCF clearly ameliorated inflammation and oxidative stress as well as pancreas and liver damages in the diabetic mice. Mechanistically, TSCF potently regulated insulin receptor (INSR)-, glucose transporter 4 (GLUT4)-, phosphatidylinositol 3-kinase (PI3K)-, and protein kinase B (PKB/AKT)-associated signaling pathways. Thus, our data collectively demonstrated that TSCF could be a promising functional food ingredient for diabetes improvement.

## 1. Introduction

Corni Fructus (CF), as a traditional Chinese medicine and food homologous plant, possesses a wide spectrum of biological activities including anti-hyperglycemia, antineoplastic, antimicrobial, anti-aging and antioxidant properties, as well as hepatoprotective and immune regulation effects [1,2,3]. As a result, CF has been extensively used in fruit wine, vinegar, jam, and health drinks to meet the increasing demand for production of functional foods that claim to provide health benefits [4,5]. CF is rich in morroniside, polyphenols, loganin, ursolic acid, iridoid glycosides, gallic acid and vitamin C [6,7]. Accumulating evidence has demonstrated that these bioactive constituents of CF could improve metabolic disorders in diabetes [8,9]. Therefore, CF has beneficial effects on treating and preventing diabetes and its complications.

Saponins, also referred to as triterpene glycosides, are a subclass of terpenoids. The amphipathic nature of saponins supplies them with the property as surfactants that can be used for the development of food, cosmetics and drugs [10]. There is enormous, commercially driven promotion of saponins as dietary supplements and functional foods. For example, the activity of β-galactosidase, one of the most important and classical biotechnological enzymes used in the food industry, can be increased by Quillaja bark saponin [10]. It is well known that saponins exhibit antidiabetic activity. Total saponins isolated from *Aralia taibaiensis* alleviated polydipsia, polyuria, polyphagia and weight loss of diabetic rats in a dose-dependent manner through increasing the levels of serum insulin, superoxide dismutase and reducing glutathione [11]. The hypoglycemic mechanism of total saponins from *Schisandra chinensis (Turcz.) Baill* was at least partially due to the activation of GLUT4, which is regulated by the IRS-1/PI3K/AKT pathway [12]. Astragalus total saponins and curcumin can achieve significant protective effects on diabetic nephropathy rats by improving the glycometabolism, insulin resistance (IR), lipid metabolism, oxidative stress levels, and pathological changes [13]. However, extremely few studies reported the isolation of the total saponins from CF (TSCF), and the role of TSCF on diabetes is almost completely unknown [14,15]. Thus, it is of great significance to study the extraction of TSCF and uncover its role in T2DM.

Unfortunately, the research on CF mainly focuses on the extraction technology of iridoid glycosides, polysaccharides, triterpenoid acids and tannins, and their anti-inflammatory and thirst-quenching properties, while the extraction technology and pharmacological effects of TSCF are rarely reported. Currently, commercially available saponins-containing products are very limited and predominantly derived from *Saponaria officinalis* and *Quillaja saponaria* by using ethanol, macroporous adsorption resin or n-butanol for extraction and polished purification. For example, the *Sanguisorba officinalis* ethanol extract, which mainly contained saponins, can inhibit bacterial toxin production at low concentrations [16]. However, the saponins in CF are more complicated and at least composed of eight glycosides [17]. In addition, the content of TS in CF is lower than that in *Saponaria officinalis* [18,19]. As a result, although various techniques including ethanol extraction, membrane separation [20], ultra-high pressure extraction [21], and chromatography were used to extract and purify TS from CF, extraction of TSCF with high yield is still difficult to achieve.

Compared to conventional extraction methods, which usually mean longer extraction time, higher temperature but with lower extraction efficiency, ultrasonic microwave-assisted extraction (UMAE), a technology that utilizes the advantages of microwave and sonochemistry to optimize the rate and efficiency of extraction, has been extensively used in the preparation of active ingredients of Chinese herbal medicine. It works well under mild conditions and thus, avoids the destruction of the components of interest [22]. As a statistical method to solve multivariable problems with high precision of regression equation and favorable fitting accuracy, response surface methodology (RSM) has been used in food, natural medicine and other fields [23]. These two methods can be combined to optimize the extraction technique with higher yields. We have reported recently that the total triterpenoid acids in CF can be extracted with high yields by UMAE optimized by RSM [24]. We wonder whether UAME can be used to extract TSCF with higher efficiency.

In this study, UMAE optimized by RSM was employed to extract TSCF, which was further purified with HPD-300 macroporous resin. By means of which the yield of TS elevated to 13.96 mg/g compared to 10.87 mg/g with unassisted extraction. In addition, we established T2DM model via feeding mice with high-fat diet (HFD) followed by streptozotocin (STZ) administration and assessed the therapeutic effects of TSCF on glucose metabolism, lipid metabolism, inflammation and oxidative stress in the diabetic mice.

## 2. Materials and Methods

### 2.1. Plant Materials

The mature fruit of CF was collected from Foping, Shaanxi Province (longitude 107.41° E to 108.10° E, latitude 33.16° N to 33.45° N, elevation 1120 m) in October 2017 and approved by Yaping Xiao, a plant identification expert from Shaanxi Normal University. CF was treated as previously described [24]. In brief, after removing the inner ripe core, the outer pulp of CF was dried, and the sarcocarp was crushed and passed through a 40-mesh sieve. The powder was stored at −20 °C for later use.

### 2.2. Chemicals and Reagents

Oleanolic acid standard (Purity ≥ 98%) was purchased from Chengdu Manster Biotechnology Co. (Chengdu, China). Streptozotocin (STZ) was from Sigma Chemical Co. (St. Louis, MO, USA). HPD-300 macroporous resin was from Shaanxi Shenlan Technology Co. Ltd. (Hanzhong, China). All the other chemicals used in the present work were at least of analytical grade.

### 2.3. Standard Curve of TS

As TS is a multi-ingredient mixture, oleanolic acid (OA) is used to draw the standard curve for TS quantification according to the method previously described [24]. Briefly, 0.2 mg/mL OA standard solution was firstly prepared by dissolving 2.0 mg OA standard substance in 10 mL methanol. Then, the solution was divided into a series of different volumes in the test tubes (concentration range: 0.2–1.0 μg/mL). After removing the methanol via rotary evaporation, 0.2 mL vanillin–glacial acetic acid solution (5%) and 0.8 mL perchloric acid solution were sequentially added to OA samples followed by a 15-min water bath at 70 °C and cooled to room temperature in flowing water. Finally, 5 mL glacial acetic acid solution was added and the absorbance value of each sample was detected at 554 nm with an ultra-micro spectrophotometer (Thermo Fisher Scientific, Waltham, MA, USA). The standard curve was drawn with the quantity of OA (Y = abscissa) and absorbance value (X = ordinate). The regression line for OA was Y = 0.0379X − 0.0209 (*R*^2^ = 0.9992).

### 2.4. Single Factor Experimental Design for Extraction of Total Saponins from CF (TSCF) with Ultrasonic-Microwave Assisted Extraction (UMAE)

The values of each single factor in the TSCF extraction process were screened using an XO-200 UMAE system by changing one single factor and keeping the others unchanged. In brief, one gram of CF powder was sealed with 70% aqueous ethanol with solid/liquid ratios of 1:10, 1:15, 1:20, 1:25, and 1:30 g/mL, respectively. The extraction time was set for 3–15 min. The ultrasonic powers ranged in 240–480 W, while the microwave powers ranged in 200–600 W. After extracting at room temperature for the indicated time, samples were centrifuged and the supernatant was gathered for absorbance value measurement as mentioned in Section 2.3. TSCF content was finally calculated with the equation:(1)$$\mathrm{content}\;\mathrm{of}\;\mathrm{TSCF}\;(\mathrm{mg}/g)=\frac{CeVe}{Me}$$
in which Ce (mg/mL), Ve (mL), and Me (g) are TSCF concentration in the extract, volume of the extract, and weight of the dried CF powder, respectively.

### 2.5. Optimization of TSCF Extraction by RSM

To optimize the extraction parameters by RSM, a four-variable *(*$X_{1}$, $X_{2}$, $X_{3}$ and $X_{4}$) three-level (high (+1), intermediate (0), and low (−1), respectively) Box–Behnken design was employed to determine the collective influence of them on the extraction of TSCF. As shown in Table 1, $X_{1}$ (solid/liquid ratio), $X_{2}$ (microwave power), $X_{3}$ (ultrasonic power), and $X_{4}$ (extraction time) are the coded variables. Thus, 29 experimental conditions were obtained and each of them was conducted in triplicate. All data were fitted to the following quadratic second degree polynomial model:(2)$$Y=\beta_{0}+\overset{4}{\underset{i=1}{\sum }}\beta_{ii}X_{i}{}^{2}+\overset{4}{\underset{i<j=2}{\sum }}\beta_{ij}X_{i}X_{j}$$
where Y is the predicted response; $\beta_{0}$ is the model constant; $\beta_{ii}$ and $\beta_{ij}$ are the quadratic and cross-product coefficients; $X_{i}$ and $X_{j}$ are independent variables. The experimental design analysis and prediction were conducted by using Design-Expert^®^ software (8.0.5b version).

### 2.6. Purification of Crude TSCF

The crude TSCF extracts were further purified by chromatography. In brief, the sample was concentrated and re-suspended with H_2_O to 3.5 mg/mL and loaded onto a column packaged with HPD-300 resin and pre-equilibrated with distilled water at a flow rate of 2 bed volume (BV)/h. After removing the unbound remnants with 3 BV of distilled water, TS were finally eluted with 8.5 BV of 50% ethanol at a flow rate of 1.5 mL/min.

To determine the TS extracted from CF, the extracted products were dissolved in aqueous ethanol to a serial-diluted concentration so that it fell within the linear range of the standard curve. Then, 0.2 mL extract solution was used for TS determination by the method in Section 2.3.

### 2.7. Establishment of T2DM Model in Mice

Sixty male ICR mice (5 weeks old) weighed 16 ± 2 g were purchased from the Animal Center of College of Medicine, Xi’an Jiaotong University (Xi’an, China). All mice were kept under controlled temperatures (22–23 °C) and relative humidity (40–60%) on a 12 h light/dark cycle and with free access to water and normal diet for 7 days before the experiment. Animal studies were carried out according to the National Institutes of Health’s Guide for the Care and Use of Laboratory Animals and were approved by the Animal Care and Use Committee of Shaanxi Normal University.

The T2DM mouse model was established according to the method we described previously [24]. In brief, the mice of the normal control (NC) group were fed with normal chow diet (11% kcal fat), whereas all the others were given a high-fat diet (HFD) (58% kcal fat, Slac Laboratory Animal, Shanghai, China) for 4 weeks. After fasting for 12 h, all HFD-fed mice received intraperitoneally administered STZ at a dosage of 60 mg/kg once every three days, four times, and equal volume of citrate buffer was administered for NC mice. Then, the mice were fasted for another 12 h and blood samples were promptly collected from the tail vein for the determination of fasting blood glucose (FBG) values. Mice with FBG ≥ 7.8 mmol/L were confirmed as T2DM for further studies.

### 2.8. Treat T2DM Mice with TSCF

The confirmed diabetic mice were divided into 5 groups: T2DM model control (DM), positive control (PC) and three TSCF-treated groups. The experimental design flow chart is shown in Figure 1. For TSCF treatment, diabetic mice were intragastrically (i.g.) administered with 50 (TSCF-L), 100 (TSCF-M), and 200 mg/kg (TSCF-H) of TSCF once a day for 4 weeks. Body weight was measured once a week. All mice were fasted for 12 h after the last administration, and blood samples were collected from the retro-orbital sinus for serum preparation and biochemical analysis. Then, the mice were killed and tissues (liver, kidney, and spleen) were immediately removed and weighed. One part of the tissues was stored at −80 °C for quantitative PCR analysis, while the other was fixed in 4% paraformaldehyde solution for histopathological examination. The organ index was calculated as the organ weight divided by the body weight.

### 2.9. Fasting Blood Glucose (FBG) and Oral Glucose Tolerance Test (OGTT)

The blood samples were obtained weekly from the tail vein of the mice after a 12-h fast, and FBG was measured by using a One-Touch glucometer (HMD Biotechnology Co., Ltd., Qingdao, China). OGTT was performed after four weeks of TSCF administration. In brief, the mice were firstly fasted for 12 h and FBG was determined as the 0 min blood glucose level. Then, glucose solution (2 g/kg) was administered to the mice by oral gavage and blood glucose (BG) values were determined at 30, 60, and 120 min, respectively. The OGTT, presented via a complete area under the curve (AUC) from 0 to 120 min, was calculated by the formula: AUC = (FBG + BG_30min_) × 1/4 + (BG_30min_ + BG_60min_) × 1/4 + (BG_60min_ + BG_120min_) × 1/2.

### 2.10. Biochemical Assay

The plasma total cholesterol (TC), total triglyceride (TG), low-density lipoprotein cholesterol (LDL-C) and high-density lipoprotein cholesterol (HDL-C), and the superoxide dismutase (SOD) activity and malondialdehyde (MDA) content of liver homogenate were measured using marketing diagnostic kits (Nanjing Jiancheng Bioengineering Institute, Nanjing, China). The inflammatory cytokines tumor necrosis factor (TNF)-α, interleukin (IL)-6, and C-reactive protein (CRP) in the liver homogenate, and fasting insulin (FINS) levels in the serum, were tested by ELISA kits (Tianjin Anoric Biotechnology Co., Ltd., Tianjin, China). All measurements were performed according to the manufacturer’s protocols. The homeostasis model assessment (HOMA)-insulin sensitivity (HOMA-IS) index and HOMA-insulin resistance (HOMA-IR) index were calculated following the formulas: HOMA-IS = ln [1/(FINS × FBG)], HOMA-IR = (FBG × FINS)/22.5 [25].

### 2.11. Histopathological Examinations

Histology of the livers and pancreas was studied using hematoxylin and eosin (H&E) staining following the standard method. In brief, fresh isolated liver and pancreas samples were sequentially fixed in 4% paraformaldehyde solution, embedded in paraffin, and cut into 5-μm thick sections. Sections were stained with H&E and were detected by light microscopy. Finally, the images were examined and evaluated with a CX23 biomicroscope (Olympus, Kyoto, Japan).

### 2.12. RNA Quantitation by Real-Time qPCR

Real-time PCR analysis of AKT, GLUT4 (glucose transport 4), INSR (insulin receptor), and PI3K mRNA expression was performed using an ABI 7500 PCR System (Applied Biosystems, Carlsbad, CA, USA). Primers for AKT were: forward 5′-TT TGGGAA GGTGATTCTGGTG-3′; reverse 5′-CGTAAGGAAGGGATGCCTAGA GTT-3′. Primers for INSR were: forward 5′-CAAGAAATGATTCAGATGACAGCAG-3′; reverse 5′-AGA CTCCATCCTTCAGGGACTCA-3′. Primers for GLUT4 were: forward 5′-CCCCATTCCC TGGTTCATT-3′; reverse 5′-GACCCATAGCATCCGCAAC-3′. Primers for PI3K were: forward 5′-GACCAATACTTGATGTGGCTGACG-3′; reverse 5′-CTCGCAATAGGTTCT CCGCTTT-3′. Primers for the control housekeeping gene β-ACTIN were: forward 5′-GC CTTCCTTCTTGGGTAT-3′; reverse 5′-GGCATAGAGGTCTTTACGG-3′. The fold change of target genes was calculated with the 2^–ΔΔCt^ method.

### 2.13. Statistical Analysis

All experiments were repeated in triplicate and the data were expressed as means ± standard deviation. Statistical analysis was performed by one-way analysis of variance (ANOVA) and Duncan’s multiple range tests and *p* < 0.05 was considered statistically significant.

## 3. Results and Discussion

### 3.1. Optimization of UMAE Conditions by RSM for TSCF Extraction

To determine which factor in UMAE mainly influences TSCF yield and the range for RSM optimization, variable factors such as solid/liquid ratio, etc., were screened by single factor experiments. As shown in Figure 2, although these factors have different effects on TSCF yield, they do have the optimal value: solid/liquid ratio = 1:20 g/mL, ultrasonic power = 360 W, extraction time = 6 min, and microwave power = 400 W. The effect of ethanol concentration on TSCF extraction was also investigated, but no apparent effects were found (data not shown). Therefore, the constant 70% ethanol was used for all experiments. Interestingly, the origin of CF has a much significant effect on TSCF extraction because of various TS contents. We previously screened CF from 15 origins and found that the TS content in Foping CF is about 25.81 ± 1.33 mg/g, which is an ideal material for TSCF extraction (unpublished Chinese article).

Then, we confirmed the most influential factors and their possible interactions via optimizing the UMAE conditions by RSM. In order to avoid losing the optimal value, a wider range including the best condition was selected for further RSM studies in which 29 experiments were designed involving three levels of four solitary factors (Table 1). The maximum yield (13.94 mg/g) of TSCF was found in the condition of $X_{1}$ = 1:20 g/mL, $X_{2}$ = 400 W, $X_{3}$ = 360 W, and $X_{4}$ = 6 min. Quadratic regression analysis of the data was carried out with Design-Expert 8.05b software and fitted with the following second-order polynomial equation:(3)$$\begin{array}{l} Y=13.76+0.34X_{1}+0.25X_{2}+0.16X_{3}-1.22X_{4}+0.25X_{1}X_{2}+0.34X_{1}X_{3}\\ -0.01X_{1}X_{4}+0.42X_{2}X_{3}+0.068X_{2}X_{4}+0.21X_{3}X_{4}\\ -0.75X_{1}{}^{2}-1.02X_{2}{}^{2}\;-0.85X_{3}{}^{2}-1.49X_{4}{}^{2} \end{array}$$
where R1 is the yield of TSCF; $X_{1}$, $X_{2}$, $X_{3}$ and $X_{4}$ represent the factors of solid/liquid ratio, microwave power, ultrasonic power, and extraction time, respectively.

The statistical significance of the regression model was measured by F-test and *p*-value, and the analysis of variance (ANOVA) for the response surface quadratic model is shown in Appendix A. The high F-value, very low *p*-value, the coefficient of determination, the adjusted coefficient of determination, non-significant lack of fit, and the adequate precision collectively demonstrated that the model was significantly accurate for prediction of response within the range of experimental variables. This was further confirmed by the fact that the predicted yields of TSCF were fitted well to the actual values (Table 1).

To determine the effects of these variables, individually and in combination, on TSCF production, three-dimensional response surface plots were generated. As shown in Figure 3, the response (TSCF production) was plotted on the Z-axis against any two independent variables, while keeping other variables at a fixed optimal level. The data were well fitted into the foregoing equation, and the most favorable level of each variable was determined to be as follows: $X_{1}$ = 1:21.47 g/mL, $X_{2}$ = 417.37 W, $X_{3}$ = 368.78 W, and $X_{4}$ = 4.81 min. For ease of operation, these parameters were adjusted and confirmed by experiments to be 1:21 g/mL, 417 W, 369 W, and 5 min, respectively. Under these conditions, the actual yield of TSCF was 13.96 ± 0.14 mg/g, which was highly consistent with the theoretical value. These data demonstrated that RSM was suitable and sufficient for TSCF extraction optimization. However, as an energy consumption technology, further studies are still needed to confirm whether UMAE may bring obstacles in the future commercial development of TSCF products.

### 3.2. Purification of Crude TSCF

After UMAE, the extraction solution was filtrated and the ethanol was removed by rotary evaporation. The insoluble residue was removed by water-saturated n-butanol two-phase extraction until the n-butanol phase was colorless. Then, n-butanol was removed by evaporation under reduced pressure, and the liquid was fixed to 20 mL with 70% ethanol for TSCF quantification. The purity of the crude TSCF is only about 6.23%, which indicated that there are still a lot of impurities in the products. We found that X-5 macroporous resin is suitable for purifying total triterpenoid acids from CF, while Zhao et al. reported that HPD-300 macroporous resin had the best adsorption and desorption properties for extracting TS from CF as compared with other eight different types of macroporous resins. Then, the TSCF crude extract was dissolved in water and loaded onto an HPD-300 macroporous resin column for further polished purification. The purity of TSCF increased from 6.23% to 37.36%, which was six times higher than before. The eluted TSCF was finally lyophilized and stored at −20 °C for later use.

At present, there is no legal standard substance for TSCF quantification. Considering the single saponin (such as ginsenoside) and the aglycone (oleanolic acid) share similar structure to TSCF, they are usually selected to draw the standard curve, and TSCF can be calculated by colorimetry. The reaction conditions should be strictly controlled because the reaction of TSCF with vanillin glacial acetic acid reagent is sensitive to temperature and time.

### 3.3. The Influence of TSCF on Body Weight and Organ Index of Diabetic Mice

We wondered whether TSCF has protective effects on diabetic mice; therefore, TSCF was used to treat T2DM mice established by HFD-feeding and STZ injection. There was no significant difference in the initial body weight of the mice in each group before TSCF administration. At the end of the study, the weight of DM mice was significantly lower than that of the NC group (*p* < 0.05). However, after 4 weeks of gavage administration of TSCF, the body weight loss of diabetic mice was significantly ameliorated and showed a relatively stable state, in contrast to the steady loss observed in the DM group (*p* < 0.05), whereas no significant difference was found between TSCF-treated and NC groups (Table 2).

The organ index can be used to determine the health of the internal organs of T2DM mice. As shown in Figure 4, kidney, liver and spleen indexes of mice in the DM group significantly increased when compared with the NC mice (*p* < 0.05 or *p* < 0.01). Actually, the liver and spleen indexes of the DM mice were nearly double those of the NC mice, which indicated that diabetes seriously damaged the liver and kidneys. However, TSCF potently suppressed these trends and all the organ indexes were restored to the normal level. In particular, no obvious differences can be found for liver, kidney and spleen indexes between TSCF-H and NC groups (*p* > 0.05). These data suggested that TSCF could significantly lessen the damage of organs in diabetic mice.

### 3.4. The Effects of TSCF on Glucose Metabolism of T2DM Mice

Impaired glycemic control is a main feature of diabetes. FBG and OGTT are the most commonly used indexes to detect diabetes mellitus, which reflects the function of islet β cells, and the ability of the body to regulate blood glucose. As shown in Figure 5A, FBG of T2DM mice elevated remarkably as compared with NC mice (*p* < 0.01). Both metformin and TSCF administration potently reduced FBG levels in T2DM mice (*p* < 0.01), although the levels were still higher than those of NC mice. In addition, the effect of high dosage of TSCF on FBG reduction was even better than metformin. As to OGTT, the blood glucose of all mice reached peak level at 30 min following oral glucose challenge. Thereafter, metformin, intermediate and high dosage of TSCF significantly suppressed the increase in blood glucose, and reached a much lower level at 120 min when compared with the DM mice (Figure 5B). Consistently, both TSCF-H and PC mice had the most significantly low levels of AUC as compared with the DM mice (Figure 5C, *p* < 0.01). These data revealed that TSCF is beneficial for blood glucose maintenance.

IR, which is indispensable for the development of T2DM, constitutes a pathophysiological state, where insulin fails to regulate glucose homeostasis in peripheral tissues [26]. In order to quantify the IR and β-cell function in diabetic mice, FINS, HOMA-IS, and HOMA-IR levels were determined. We found that FINS and HOMA-IS were downregulated, whereas HOMA-IR was upregulated in the mice of the DM group as compared with the mice of the NC group (*p* < 0.01) (Table 3), which indicated damaged islet cells and obvious IR. However, this condition was greatly relieved by TSCF treatment and the FINS level was clearly restored to the normal level in the TSCF-M and TSCF-H groups; there were no significant differences between them and the mice of NC group (*p* > 0.05). Similarly, TSCF clearly improved the insulin sensitivity in T2DM mice as indicated by the HOMA-IS and HOMA-IR levels (*p* < 0.05 or *p* < 0.01 vs. DM group), and all of them were restored to the normal baseline under TSCF treatment. Notably, the effect of TSCF-H was comparable to, if not more potent than, that of the positive metformin. Collectively, all these data confirmed that TSCF could increase insulin secretion and sensitivity and relieve IR.

### 3.5. The Effects of TSCF on Lipid Metabolism in T2DM Mice

The metabolic profile of T2DM includes not only impaired glucose metabolism, but also dyslipidemia combined with elevated lipid profile levels [27]. Table 4 shows the effect of TSCF on the TG, TC, LDL-C and HDL-C levels in diabetic mice. Compared with NC mice, DM mice exhibited significantly higher levels of TG, TC and LDL-C; lower levels of HDL-C. TSCF treatment for 4 weeks led to a great decrease in TG and LDL-C (*p* < 0.01) and a significant increase in HDL-C (*p* < 0.01) as compared with the mice of the DM group. A number of recent studies have claimed that diabetes is strongly associated with increased lipid accumulation, and elevated blood glucose was accompanied by increases in serum TC and TG in diabetic rats [28]. The interesting findings of this study are that treatment with TSCF reduced not only FBG but also total cholesterol and TG levels in T2DM mice. At the same time, administration of TSCF increased HDL-C but decreased LDL-C. These results revealed that TSCF could improve lipid metabolism in T2DM mice.

### 3.6. Effects of TSCF on Inflammation and Oxidative Stress in T2DM Mice

Inflammation is the body’s defensive response to stimulation, and inflammatory factors play critical roles in regulating the insulin signaling pathway and the structure and function of islet β cells [29]. IL-6, TNF-α, CRP and other inflammatory factors secreted by various cells can induce a cytokine signal to activate c-Jun amino terminal kinase (JNK), reducing the activation of downstream PI3K, thus inhibiting the expression and synthesis of GLUT4, and leading to β cell dysfunction and IR [30,31]. We found that IL-6, TNF-α, and CRP levels were greatly upregulated in the liver tissues of T2DM mice compared to the NC mice. After treating these T2DM mice with TSCF for 4 weeks, all these cytokines were decreased significantly (*p* < 0.01) in a TSCF dose-dependent manner (Figure 6A–C).

Considering the levels of MDA and SOD in liver tissues reflect the severity of free radical attack and the ability of scavenging free radicals, they can be used to detect the damage of liver function. As shown in Figure 6D,E, MDA clearly increased, whereas SOD activities significantly decreased in the diabetic mice as compared with the normal mice (*p* < 0.01). In contrast, both MDA levels and SOD activities tended to restore to normal levels as compared to DM mice after a 4-week treatment with TSCF. Additionally, TSCF dose-dependently recovered SOD levels of the T2DM mice. It has been reported that hepatic inflammation and oxidative stress are closely connected with diabetes [32]. Thus, all these data suggested that TSCF potently attenuated inflammation and oxidative stress in T2DM mice.

### 3.7. TSCF Ameliorated Pancreas and Liver Damages in T2DM Mice

The trends of organ index, glucose metabolism, inflammatory cytokines production and oxidative stress suggested that TSCF could improve organ damages in T2DM mice. To confirm this, histopathological assays were performed. As shown in Figure 7, HFD and STZ triggered severe injury to the pancreas and liver of the diabetic mice. Intact pancreatic structure and neatly aligned islet β cells can be seen in NC mice, while the pancreatic cells were severely damaged, with dilated ducts and accumulated fat tissue, represented by big and optically empty cells filled with lipids, in the DM and TSCF-L groups. However, all these features were significantly reduced in TSCF-M, TSCF-H and PC groups (Figure 7A). As to liver tissues, NC mice showed intact normal hepatic histology, but HFD-STZ-induced DM mice exhibited numerous cytotoxic and inflammatory alterations including steatosis with some affected cells, vacuolar and hydropic degenerations, hepatocellular necrosis, and leukocyte infiltrations. In contrast, a significant decline in the severity and frequency of hepatic lesions was observed in the livers of TSCF-treated mice (Figure 7B). Meanwhile, the protective effect of TSCF on liver can also be found by organ index (Figure 4A). All these data revealed that TSCF could ameliorate the hepatic and pancreatic injuries in T2DM mice, and therefore, may be a promising candidate for diabetes treatment.

### 3.8. TSCF Regulates INSR, GLUT4, PI3K and AKT Signaling Pathways in Skeletal Muscle of T2DM Mice

It is well known that skeletal muscle is the principal site for glucose regulation and metabolism, and muscle insulin resistance is pivotal to abnormal glucose metabolism in diabetes. Considering that INSR-, GLUT4-, PI3K- and AKT-associated signaling pathways play critical roles in insulin-glucose regulation and homeostasis in skeletal muscle, we determined the effects of high-dosage TSCF on these signaling pathways to elucidate the possible molecular mechanisms of TSCF on T2DM mice. We found that T2DM induction greatly decreased (*p* < 0.01), whereas high-dose TSCF and metformin administration remarkably restored the level of INSR, GLUT4, PI3K and AKT in skeletal muscle of mice (Figure 8). In addition, the effect of TSCF-H on the expression of GLUT4 in the skeletal muscle of mice was seven times greater than that of the metformin (*p* < 0.05). These data suggested that TSCF suppressed hyperlipidemia and hyperglycemia in diabetic mice by regulating the INSR-, GLUT4-, PI3K- and AKT-associated signaling pathways to maintain the homeostasis of insulin-glucose metabolism.

## 4. Conclusions

T2DM is a complex metabolic disorder associated with pancreas dysfunction and varying degrees of insulin resistance, accounting for over 90% of all individuals diagnosed with diabetes. CF has great potential as a drug homologous food for the treatment of diabetes. Optimizing the processing technology is an effective way to improve the yield of active ingredients for the industrial production of medicinal and food homologous plants. In this study, RSM was used to optimize the UMAE of TSCF and the yield of TSCF increased from 10.87 up to 13.96 mg/g. If administered to HFD-TSZ-induced T2DM mice, TSCF significantly ameliorated the syndrome of diabetic mice as indicated by improved glucose and lipid metabolism, reduced inflammation and oxidative stress, and ameliorated pancreas and liver damages in the mice. Mechanistically, TSCF may alleviate hyperlipidemia and hyperglycemia in diabetic mice by regulating the INSR-, GLUT4-, PI3K- and AKT-associated signaling pathways. In summary, TSCF has a therapeutic effect on T2DM and can be used as a potential natural product to treat diabetes.

## Figures and Tables

**Figure 1 foods-10-00670-f001:**
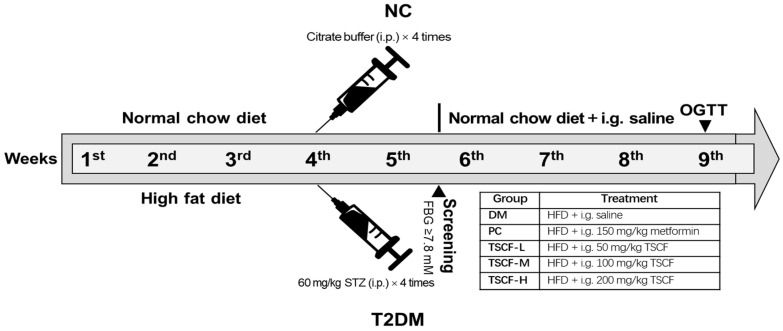
Schematic diagram of the experimental process. i.p., intraperitoneal injection; i.g., intragastric administration; HFD, high-fat diet; OGTT, oral glucose tolerance test; FBG, fasting blood glucose; STZ, streptozotocin; NC, normal control group; DM, diabetes mellitus model group; PC, positive (metformin-treated) control group; TSCF, total saponins from Corni Fructus; TSCF-L, low-dose TSCF group; TSCF-M, middle-dose TSCF group; TSCF-H, high-dose TSCF group.

**Figure 2 foods-10-00670-f002:**
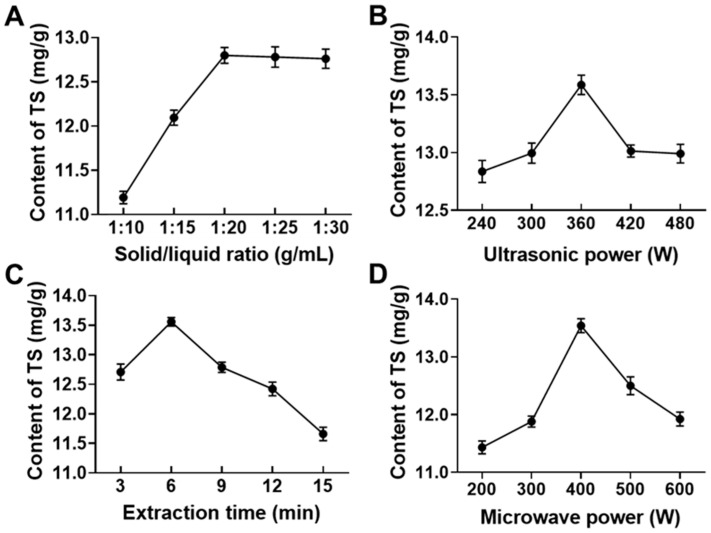
Screening of the variable factors that affect TSCF extraction yield. TSCF was extracted under certain conditions, including solid/liquid ratio (**A**), ultrasonic power (**B**), microwave power (**C**), and extraction time (**D**), and the extraction yield was determined.

**Figure 3 foods-10-00670-f003:**
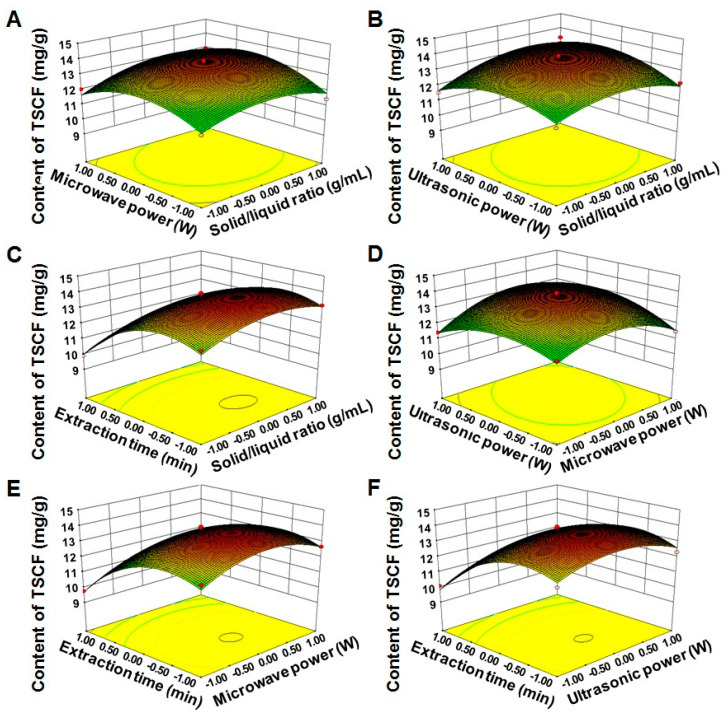
Three-dimensional response surface plot for TSCF production. The contour plots showed the interactive effects of the solid/liquid ratio and microwave power (**A**), solid/liquid ratio and ultrasonic power (**B**), solid/liquid ratio and extraction time (**C**), microwave power and ultrasonic power (**D**), microwave power and extraction time (**E**), and ultrasonic power and extraction time (**F**), respectively.

**Figure 4 foods-10-00670-f004:**
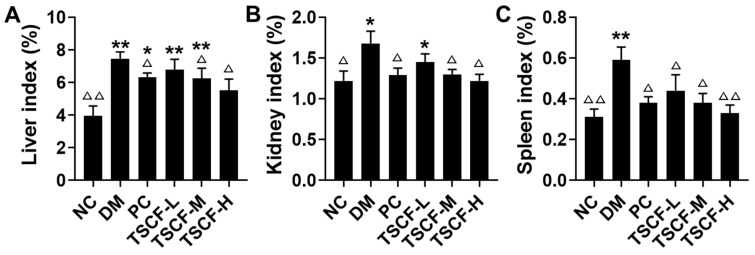
Effects of TSCF on the organ indexes of diabetic mice. (**A**) Liver index; (**B**) kidney index; (**C**) spleen index. NC, normal control group; DM, diabetic mice treated with PBS; PC, diabetic mice treated with 100 mg/kg metformin each day; TSCF-L, TSCF-M, and TSCF-H, diabetic mice treated with 50, 100, and 200 mg/kg TSCF each day. * *p* < 0.05, ** *p* < 0.01 compared with the NC group; ^Δ^
*p* < 0.05, ^ΔΔ^
*p* < 0.01, compared with the DM group.

**Figure 5 foods-10-00670-f005:**
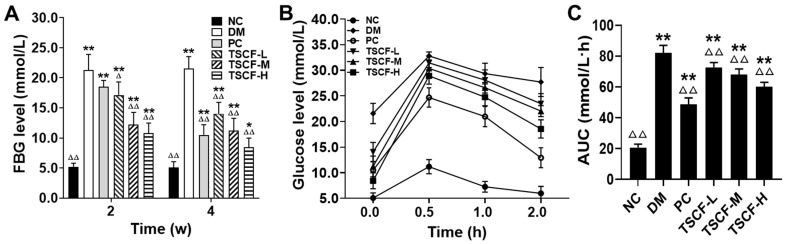
Effects of TSCF on glycemic modulation of T2DM mice. (**A**) FBG; (**B**) OGTT; (**C**) AUC; * *p* < 0.05, ** *p* < 0.01 compared with the NC group; ^Δ^
*p* < 0.05, ^ΔΔ^
*p* < 0.01 compared with the DM group.

**Figure 6 foods-10-00670-f006:**
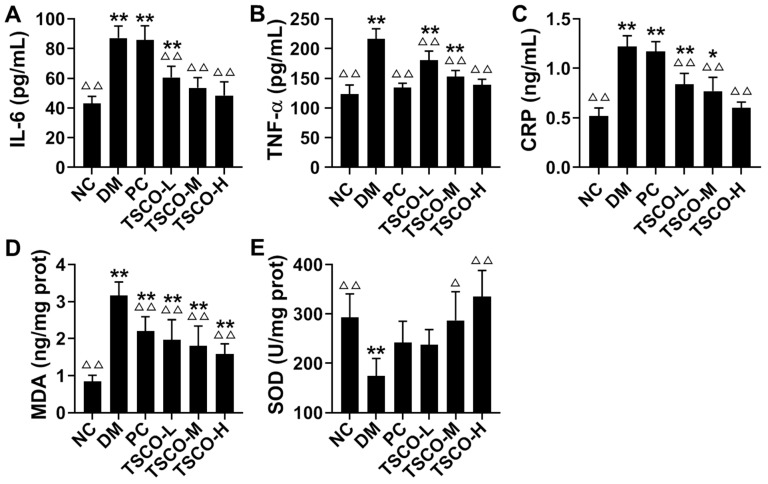
Effects of TSCF on inflammation and oxidative stress in T2DM mice. Sera of diabetic mice were prepared and (**A**) IL-6, (**B**) TNF-α, (**C**) CRP, (**D**) MDA, and (**E**) SOD levels were measured. NC, normal control group; DM, T2DM model group; PC, Metformin group; TSCF-L, TSCF-M, and TSCF-H, diabetes mice treated with 50 mg/kg, 100 mg/kg and 200 mg/kg TSCF per day, respectively. * *p* < 0.05, ** *p* < 0.01 compared with NC group; ^Δ^
*p* < 0.05, ^ΔΔ^
*p* < 0.01 compared with DM group.

**Figure 7 foods-10-00670-f007:**
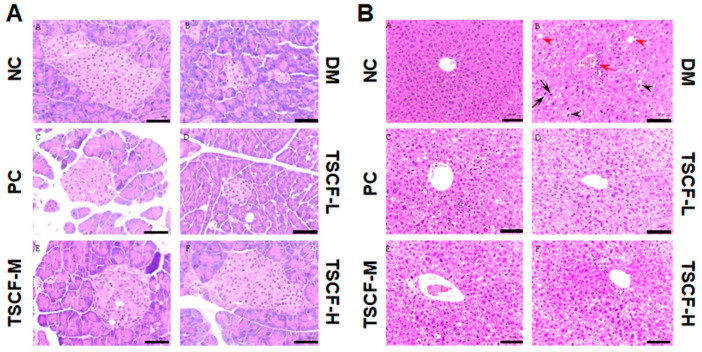
Histopathological changes of diabetic mice treated with TSCF. Pancreas (**A**) and liver (**B**) tissues were isolated from diabetic mice and H&E staining was performed according to standard methods. Apoptotic hepatocytes (black arrow), focal mononuclear cell aggregation (red arrow), diffuse hydropic degeneration (black arrowheads), fatty change (red arrowheads). NC, normal control group; DM, T2DM model group; PC, Metformin group; TSCF-L, TSCF-M, and TSCF-H, diabetes mice treated with 50, 100 and 200 mg/kg TSCF per day, respectively. Bar = 100 μm.

**Figure 8 foods-10-00670-f008:**
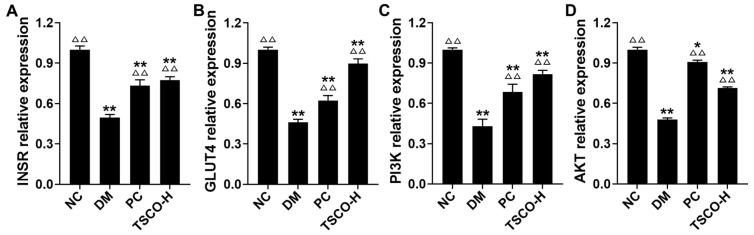
The effects of TSCF on MDA (**A**) INSR, (**B**) GLUT4, (**C**) PI3K, and (**D**) AKT signaling pathways in T2DM mice. * *p* < 0.05, ** *p* < 0.01 compared with NC group; ^△^
*p* < 0.05, ^△△^
*p* < 0.01 compared with DM group.

**Table 1 foods-10-00670-t001:** Results of central composite experimentation.

No.	*X*_1_ (g/mL)	*X*_2_ (W)	*X*_3_ (W)	*X*_4_ (min)	Theoretical Yield of TS (mg/g)	Actual Yield of TS (mg/g)
1	0 (1/20)	−1 (300	1 (420)	0 (6)	11.38	11.42
2	0 (1/20)	1 (500)	−1 (300)	0 (6)	11.56	11.48
3	0 (1/20)	1 (500)	0 (360)	−1 (3)	12.66	12.69
4	1 (1/25)	0 (400)	0 (360)	1 (9)	10.64	10.38
5	0 (1/20)	−1 (300)	0 (360)	1 (9)	9.71	9.76
6	−1 (1/15)	−1 (300)	0 (360)	0 (6)	11.65	11.54
7	0 (1/20)	0 (400)	1 (420)	−1 (3)	12.60	12.31
8	0 (1/ 20)	0 (400)	0 (360)	0 (6)	13.76	13.94
9	0 (1/ 20)	0 (400)	0 (360)	0 (6)	13.76	13.75
10	0 (1/ 20)	0 (400)	−1 (300)	1 (9)	9.83	10.08
11	0 (1/ 20)	−1 (300)	0 (360)	−1 (3)	12.29	12.57
12	−1 (1/15)	1 (500)	0 (360)	0 (6)	11.65	12.02
13	−1 (1/15)	0 (400)	1 (420)	0 (6)	11.65	11.52
14	1 (1/ 25)	1 (500)	0 (360)	0 (6)	12.83	12.91
15	0 (1/ 20)	0 (400)	−1 (300)	−1 (3)	12.70	12.42
16	0 (1/ 20)	0 (400)	1 (420)	1 (9)	10.58	10.82
17	0 (1/ 20)	1 (500)	1 (420)	0 (6)	12.71	12.51
18	0 (1/ 20)	0 (400)	0 (360)	0 (6)	13.76	13.57
19	0 (1/ 20)	0 (400)	0 (360)	0 (6)	13.76	13.85
20	0 (1/ 20)	0 (400)	0 (360)	0 (6)	13.76	13.71
21	1 (1/ 25)	0 (400)	1 (420)	0 (6)	13.01	13.34
22	0 (1/ 20)	−1 (300)	−1 (300)	0 (6)	11.89	12.05
23	1 (1/ 25)	0 (400)	−1 (300)	0 (6)	12.00	12.21
24	−1 (1/15)	0 (400)	−1 (300)	0 (6)	12.00	11.75
25	−1 (1/15)	0 (400)	0 (360)	1 (9)	9.98	9.89
26	0 (1/ 20)	1 (500)	0 (360)	1 (9)	10.35	10.15
27	1 (1/ 25)	−1 (300)	0 (360)	0 (6)	11.83	11.43
28	1 (1/ 25)	0 (400)	0 (360)	−1 (3)	13.11	13.15
29	−1 (1/15)	0 (400)	0 (360)	−1 (3)	12.40	12.62

**Table 2 foods-10-00670-t002:** Effect of TSCF on the body weight of diabetic mice.

Group	Body Weight (g)				
	0 w	1 w	2 w	3 w	4 w
NC	33.33 ± 2.78	33.68 ± 2.70	34.04 ± 1.80 ^△^	33.66 ± 1.85 ^△^	32.71 ± 1.63 ^△^
DM	33.29 ± 2.69	31.35 ± 2.76	30.89 ± 1.89 *	29.01 ± 2.13 **	28.65 ± 1.48 *
PC	32.81 ± 1.74	32.18 ± 2.35	32.41 ± 2.28	32.73 ± 2.32 ^△^	32.42 ± 1.56 ^△^
TSCF-L	33.23 ± 1.98	32.04 ± 2.28	32.47 ± 0.81	32.47 ± 0.81 ^△^	32.15 ± 1.57 ^△^
TSCF-M	33.65 ± 2.67	33.07 ± 2.65	32.54 ± 1.46	33.26 ± 2.22 ^△△^	32.41 ± 1.17 ^△^
TSCF-H	32.48 ± 2.38	32.58 ± 1.66	32.27 ± 1.10	33.33 ± 1.77 ^△△^	32.25 ± 1.63 ^△^

Data were expressed as means ± SD (*n* = 10). * *p* < 0.05, ** *p* < 0.01 vs. NC group; ^△^
*p* < 0.05, ^△△^
*p* < 0.01 vs. DM group.

**Table 3 foods-10-00670-t003:** Effect of TSCF on FINS, HOMA-IS, and HOMA-IR of diabetic mice.

Group	FINS (mIU/L)	HOMA-IS	HOMA-IR
NC	9.46 ± 1.71 ^△△^	–3.84 ± 0.25 ^△△^	2.13 ± 0.55 ^△△^
DM	5.89 ± 0.73 **	–4.93 ± 0.15 **	6.19 ± 1.02 **
PC	8.40 ± 1.92 ^△△^	–4.46 ± 0.22 **^,△△^	3.92 ± 0.90 **^,^^△△^
TSCF-L	7.03 ± 0.97 *	–4.53 ± 0.19 **^,△△^	4.38 ± 0.61 **^,^^△△^
TSCF-M	8.26 ± 0.69 ^△△^	–4.45 ± 0.15 **^,△△^	3.84 ± 0.57 **^,^^△△^
TSCF-H	9.43 ± 1.53 ^△△^	–4.31 ± 0.08 **^,△△^	3.52 ± 0.81 *^,^^△△^

Data were expressed as means ± SD (*n* = 10). * *p* < 0.05, * *p* < 0.01 vs. NC group; ^△^
*p* < 0.05, ^△△^
*p* < 0.01 vs. DM group.

**Table 4 foods-10-00670-t004:** Effect of TSCF on the lipid levels in diabetic mice ($\bar{x}\pm s$, mmol/L).

Group	TC	TG	HDL-C	LDL-C
NC	2.49 ± 0.11 ^△△^	0.67 ± 0.08 ^△△^	1.68 ± 0.13 ^△△^	2.98 ± 0.79 ^△△^
DM	2.89 ± 0.28 **	0.93 ± 0.11 **	1.01 ± 0.96 **	4.81 ± 0.71 **
PC	2.66 ± 0.13	0.68 ± 0.11 ^△△^	1.66 ± 0.11 ^△△^	4.48 ± 0.71 **
TSCF-L	2.50 ± 0.23 ^△△^	0.73 ± 0.09 ^△△^	1.35 ± 0.09 **^,^^△△^	3.70 ± 0.30 *^,^^△△^
TSCF-M	2.69 ± 0.16	0.70 ± 0.06 ^△△^	1.39 ± 0.12 **^,^^△△^	3.45 ± 0.25 ^△△^
TSCF-H	2.68 ± 0.23	0.67 ± 0.10 ^△△^	1.46 ± 0.11 *^,^^△△^	3.21 ± 0.43 ^△△^

Data were expressed as means ± SD (*n* = 10). * *p* < 0.05, ** *p* < 0.01 compared with NC group; ^△^
*p* < 0.05, ^△△^
*p* < 0.01 compared with DM group.

## Data Availability

Not applicable.

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
