# Peer review of "Total Saponins Isolated from Corni Fructus via Ultrasonic Microwave-Assisted Extraction Attenuate Diabetes in Mice"

_foods, 2021, doi:10.3390/foods10030670_

Round 1

Reviewer 1 Report

The work foods-1127680 addresses the ultrasonic-microwave assisted extraction optimized by response surface methodology to extract saponins from Corni Fructus, which were further purified with HPD-300 macroporous resin. Finally, authors established T2DM model via feeding mice with high-fat diet followed by streptozotocin injection and assessed the therapeutic effects of TSCF on glucose metabolism, lipid metabolism, inflammation and oxidative stress in the diabetic mice. The document was written with rigor and contains a considerable amount of data that makes it quite interesting. However, there are some issues that the authors should consider.

Major concerns

Firs of all, this work presents quite obvious similarities with another work previously published by the same authors of this study: Food Funct., 2020,11, 10709-10723. Among other things, the methods are almost a carbon copy, which suggests to me that both works come from the same research. Therefore, it is the authors' duty to make it clear in the Introduction section, along with Discussions, why this work does not represent a salami publication.

- I am not entirely convinced that there has been a significant extraction yield improvement by using the ultrasonic-microwave assisted extraction technique. The authors should consider in their discussion that by using a technique that provides energy expenditure, the commercial viability of the product is compromised.

- In Section 2.4 and 2.5 factors should be specified

- In Section 3.1., before starting to optimize, I encourage authors to analyze and discuss about the actual influence of each factor on the response variable (TSCF yield, as shown in Figure 2) they are discussing. This discussion can be complemented by the statistical influences found. However, the actual results and the analysis of their actual physical factors should not be overlooked.

I consider that there is a total lack of discussion of actual experimental results, while spending much of the discussion on optimization data, which we must remember are just that, predicted data.

Minor concerns

- There is an excessive number of abbreviations. Most of them could be avoided.

- L46: check the use of capital letter.

- L42: S. chinensis had not been mentioned before, therefore, the complete name should be used. Also, this must be in italics.

- L57: Authors claim, “the role of TSCF on diabetes is almost completely unknown.” This must be supported.

- L60: Authors claimed “In general, there are two major saponin extraction methods, namely conventional and contemporary methods (also referred to as green methods)”. The word contemporary should be change. I remind the authors that there are old methods that are still green. Moreover, the ultrasonic-microwave assisted extraction method has been implemented for at least two decades.

- L70: Avoid novel.

- Table 2 should be left as a supplementary file.

Author Response

Response to Reviewer Comments

Manuscript title: Total saponins isolated from Corni Fructus via ultrasonic- microwave assisted extraction attenuates diabetes in mice

Manuscript No.: foods-1127680

Dear Editor and reviewers,

We are grateful to you for your valuable suggestions. Those comments are very helpful for revising and improving our manuscript, as well as for guiding our future researches. Based on your comments and request, we have seriously and carefully revised our manuscript. A revised version of our manuscript with the corrected sections as well as the newly added contents marked with yellow, has been attached.

Reviewer 1

The work foods-1127680 addresses the ultrasonic-microwave assisted extraction optimized by response surface methodology to extract saponins from Corni Fructus, which were further purified with HPD-300 macroporous resin. Finally, authors established T2DM model via feeding mice with high-fat diet followed by streptozotocin injection and assessed the therapeutic effects of TSCF on glucose metabolism, lipid metabolism, inflammation and oxidative stress in the diabetic mice. The document was written with rigor and contains a considerable amount of data that makes it quite interesting. However, there are some issues that the authors should consider.

Major concerns

Firs of all, this work presents quite obvious similarities with another work previously published by the same authors of this study: Food Funct., 2020, 11, 10709-10723. Among other things, the methods are almost a carbon copy, which suggests to me that both works come from the same research. Therefore, it is the authors' duty to make it clear in the Introduction section, along with Discussions, why this work does not represent a salami publication.

Response: Thank you for your suggestion. We have carefully revised the main text in the paper based on the check report provided by the editor so that the repetitive rate is low enough according to the publication rule. Although the present work seems similar to the paper we published on Food & Function, the two studies have completely different meanings. The total triterpenoid acids from Corni Fructus are clear ingredients and easily to be purified, while total saponins in many Chinese herbal medicine or healthy foods are more complicated with powerful biological functions. Thus, any slight optimization of purification technology on saponins is precious. In addition, the crude saponins extraction method seems the same, but the factors and their ranges we selected are different. For example, concentration of aqueous ethanol has significant effect on total triterpenoid acids extraction but has no effect on saponins extraction in this work. Furthermore, for polished purification, the macroporous resins selected for these two contents are different. So, we confirmed that these two papers are different and the present work is more meaningful than the former one.

 - I am not entirely convinced that there has been a significant extraction yield improvement by using the ultrasonic-microwave assisted extraction technique. The authors should consider in their discussion that by using a technique that provides energy expenditure, the commercial viability of the product is compromised.

Response: Thanks for your comment. The UMAE method has become increasingly popular on account of superiorities in some aspects including higher extraction efficiency, shorter extraction time, energy savings, and reduced by-products [1]. Ultrasonic-microwave synergistic extraction technology has the advantages of short extraction time and high extraction rate, which has been widely used in the extraction of bioactive substances [2-4]. In fact, we also want to confirm that there has been a significant extraction yield improvement by using UMAE. This is why we used UMAE method to purify different ingredients from Corni Fructus. We have added discussion at the end of the section 3.1 according to your suggestion.

  • Xu SY et al. Ultrasonic-microwave assisted extraction, characterization and biological activity of pectin from jackfruit peel. LWT 2018; 90: 577-582.
  • Qian L et al. Optimized microwave-assistant extraction combined ultrasonic pretreatment of flavonoids from Periploca forrestii Schltr and evaluation of its anti-allergic activity. Electrophoresis 2017; 38(8): 1113-1121.
  • Xu SY, Chen XQ, Liu Y et al. Ultrasonic/microwave-assisted extraction, simulated digestion, and fermentation in vitro by human intestinal flora of polysaccharides from Porphyra haitanensis. Int J Biol Macromol 2020;152: 748-756.
  • Cheng CY, Duan WW, Duan ZH et al. Extraction of Chondroitin Sulfate from Tilapia Byproducts with Ultrasonic-Microwave Synergistic. Adv Mater Res 2013; 726-731:4381-4385.

- In Section 2.4 and 2.5 factors should be specified

Response: We have carefully revised these two sections and we wish it will be helpful.

- In Section 3.1., before starting to optimize, I encourage authors to analyze and discuss about the actual influence of each factor on the response variable (TSCF yield, as shown in Figure 2) they are discussing. This discussion can be complemented by the statistical influences found. However, the actual results and the analysis of their actual physical factors should not be overlooked.

Response: We have added discussion in this section and we wish it will be helpful.

I consider that there is a total lack of discussion of actual experimental results, while spending much of the discussion on optimization data, which we must remember are just that, predicted data.

Response: Thank you for your valuable suggestion which will guide our future work. We have added some discussion about the actual experimental data and deleted some discussion about the optimization data in the manuscript.

Minor concerns

 - There is an excessive number of abbreviations. Most of them could be avoided.

Response: Thank you for your suggestion and we have deleted all the abbreviations that used a few times in the paper.

- L46: check the use of capital letter.

Response: We have corrected it.

- L42: S. chinensis had not been mentioned before, therefore, the complete name should be used. Also, this must be in italics.

Response: We have corrected it.

- L57: Authors claim, “the role of TSCF on diabetes is almost completely unknown.” This must be supported.

Response: Thanks for your suggestion. We have added some references to support us.

  • Lin MH, Liu HK, Huang WJ, Huang CC, Wu TH, Hsu FL. Evaluation of the potential hypoglycemic and Beta-cell protective constituents isolated from Corni fructus to tackle insulin-dependent diabetes mellitus. J Agric Food Chem 2011; 59:7743-7751.
  • Huang J, Zhang YW, Dong Lin et al. Ethnopharmacology, phytochemistry, and pharmacology of Cornus officinalis et Zucc. J Ethnopharmacol2018; 213:280-301.

 - L60: Authors claimed “In general, there are two major saponin extraction methods, namely conventional and contemporary methods (also referred to as green methods)”. The word contemporary should be change. I remind the authors that there are old methods that are still green. Moreover, the ultrasonic-microwave assisted extraction method has been implemented for at least two decades.

Response: We have corrected the words according to your suggestion.

- L70: Avoid novel.

Response: We have corrected it.

- Table 2 should be left as a supplementary file.

Response: We have changed the Table 2 to be a supplementary file.

Reviewer 2 Report

I have read the manuscript with interest. The topic falls in the scope of this journal. I found this manuscript very complete, detailed and well written. However, there are still some minor changes that need to be made by the authors.

My comments are detailed below.

Keywords:

It is better to avoid repetition of words already present in the title. By using words not present in the title, you increase the possibility of finding it in articles searches. I suggest to replace Corni fructus (e.g. with Cornus officinalis), saponins and ultasonic-microwave assisted extraction.

Materials and Methods:

Line 104: the concentration of OA standard solution is 0.02 mg/mL.

Line 105-106: I suggest to add the concentrations (or a range of concentrations) of the standard solutions used to perform the calibration curve.

Line 113-114: since Y=ordinate and X=abscissa, the word order in the sentence must be reversed with “as the ordinate and abscissa, respectively.”

Line 150-151: “To determine the TS…the extracted products were dissolved in ethanol to a certain concentration...”. Can you shed any more light on this sentence of yours? What concentration? Are the extracts dried and then dissolved in ethanol?

Line 151: Change "ml" with "mL".

Line 154: Table 1, which contains results, must be inserted in paragraph 3.1.

Results and Discussion:

Line 286-288 and 290: Paragraph 3.2 needs to be amended. These sentences are part of “materials and methods”. The results are in line 289 only (The purity of TSCF…than before.).

Author Response

Response to Reviewer Comments

Manuscript title: Total saponins isolated from Corni Fructus via ultrasonic- microwave assisted extraction attenuates diabetes in mice

Manuscript No.: foods-1127680

Dear Editor and reviewers,

We are grateful to you for your valuable suggestions. Those comments are very helpful for revising and improving our manuscript, as well as for guiding our future researches. Based on your comments and request, we have seriously and carefully revised our manuscript. A revised version of our manuscript with the corrected sections as well as the newly added contents marked with yellow, has been attached.

Reviewer 2

I have read the manuscript with interest. The topic falls in the scope of this journal. I found this manuscript very complete, detailed and well written. However, there are still some minor changes that need to be made by the authors.

Keywords:

It is better to avoid repetition of words already present in the title. By using words not present in the title, you increase the possibility of finding it in articles searches. I suggest to replace Corni fructus (e.g. with Cornus officinalis), saponins and ultasonic-microwave assisted extraction.

Response: Thank you for the good suggestion and we have changed the keywords.

Materials and Methods:

Line 104: the concentration of OA standard solution is 0.02 mg/mL.

Response: We have added some information to make us understood.

Line 105-106: I suggest to add the concentrations (or a range of concentrations) of the standard solutions used to perform the calibration curve.

Response: We have added the information.

Line 113-114: since Y=ordinate and X=abscissa, the word order in the sentence must be reversed with “as the ordinate and abscissa, respectively.”

Response: We have corrected it according to your suggestion.

Line 150-151: “To determine the TS…the extracted products were dissolved in ethanol to a certain concentration...”. Can you shed any more light on this sentence of yours? What concentration? Are the extracts dried and then dissolved in ethanol?

Response: Thank you for the good suggestion and we have added some information to make us understandable.

Line 151: Change "ml" with "mL".

Response: We have corrected it.

Line 154: Table 1, which contains results, must be inserted in paragraph 3.1.

Response: We have changed it according to your suggestion.

Results and Discussion:

Line 286-288 and 290: Paragraph 3.2 needs to be amended. These sentences are part of “materials and methods”. The results are in line 289 only (The purity of TSCF…than before.).

Response: We have amended the information according to your suggestion and we wish it will be helpful.

Round 2

Reviewer 1 Report

In reference to the first reply, I recommend the authors to clarify where they introduced this in the manuscript, as I recommended in my first review.

On the other hand, the authors do not specify where the changes were introduced in their response letter, which is necessary to evidence the changes.

Minor changes in the new version

L107: (5% (w/v)). 

Author Response

Reviewer 1 Round 2

In reference to the first reply, I recommend the authors to clarify where they introduced this in the manuscript, as I recommended in my first review.

Response: We have added some information in the manuscript Introduction section as you recommended, we wish it will be helpful.

On the other hand, the authors do not specify where the changes were introduced in their response letter, which is necessary to evidence the changes.

Response: Thank you so much for your kindly suggestion and we have located our corrections (page and line/paragraph numbers) in the response letter to your Round 1 comments.

Minor changes in the new version

Response: We have added some new revisions to the manuscript

L107: (5% (w/v)). 

Response: We have deleted (w/v).
